# Administration of Different Doses of Acrylamide Changed the Chemical Coding of Enteric Neurons in the Jejunum in Gilts

**DOI:** 10.3390/ijerph192114514

**Published:** 2022-11-04

**Authors:** Michał Bulc, Jarosław Całka, Katarzyna Palus

**Affiliations:** Department of Clinical Physiology, Faculty of Veterinary Medicine, University of Warmia and Mazury in Olsztyn, Oczapowski Str. 13, 10-718 Olsztyn, Poland

**Keywords:** acrylamide, jejunum, enteric nervous system, pig

## Abstract

Excessive consumption of highly processed foods, such as chips, crisps, biscuits and coffee, exposes the human to different doses of acrylamide. This chemical compound has a multidirectional, adverse effect on human and animal health, including the central and peripheral nervous systems. In this study, we examined the effect of different doses of acrylamide on the enteric nervous system (ENS) of the porcine jejunum. Namely, we took into account the quantitative changes of neurons located in the jejunum wall expressing substance P (SP), galanin (GAL), a neuronal form of nitric oxide synthase (nNOS), the vesicular acetylcholine transporter (VAChT) and cocaine- and amphetamine-regulated transcript (CART). The obtained results indicate that acrylamide causes a statistically significant increase in the number of neurons immunoreactive to SP, GAL, VAChT and CART in all types of examined enteric plexuses and a significant drop in the population of nNOS-positive enteric neurons. Changes were significantly greater in the case of a high dose of acrylamide intoxication. Our results indicate that acrylamide is not indifferent to ENS neurons. A 28-day intoxication with this substance caused marked changes in the chemical coding of ENS neurons in the porcine jejunum.

## 1. Introduction

Acrylamide (CH2 = CH–CO–NH2, prop-2-enamide) is a low-molecular-weight organic compound composed of carbon (50.69%), hydrogen (7.09%), nitrogen (19.71%), and oxygen (22.51%) atoms. It is an odorless and flavorless crystalline substance with a melting point of 84.5 °C, readily soluble in water and organic solvents, such as acetone and ethanol [1]. After entering the soil, it becomes biodegraded [2]. In the laboratory, acrylamide is used for the selective modification of sulfhydryl (–SH) groups of proteins, while polyacrylamide is used for the electrophoretic separation of proteins and DNA [3]. Interest in acrylamide increased significantly in recent years, mainly as a result of studies conducted by Swedish scientists, which indicated large amounts of this compound in food products, particularly those containing large amounts of carbohydrates and subjected to heat treatment [4]. The mechanism of acrylamide formation in food is well understood. The results of the studies published to date indicate that the factor necessary for its formation is an elevated temperature of at least up to approximately 100 °C. The main factor leading to the formation of acrylamide in food products is the Maillard reaction in which aldehydes, ketones and carbohydrates condense with amino acids, peptides and proteins. This reaction imparts the flavor and color to food products, and one of its consequences is the formation of acrylamide [1,5,6,7].

The exposure of an animal or human organism to acrylamide has an adverse effect on the central and peripheral nervous systems in particular [8,9]. This toxicity involves the impairment of the function of nervous tissue enzymes, in particular keratin kinase, and the impairment in neurotransmitter release [10]. One of the main routes by which acrylamide enters the organism is the gastrointestinal tract [11]. This leads to its absorption and systemic intoxication, while causing damage within the gastrointestinal tract itself. The components of the gastrointestinal tract that are most vulnerable to damage include the enteric nervous system [12]. This system is a characteristic feature of the gastrointestinal tract, which gives its particular sections relative functional autonomy. The enteric nervous system components are present in all structures of the gastrointestinal tract (esophagus, stomach, small and large intestines). This system regulates, via the submucosal plexus situated between the submucosal and muscle layer (divided in large animals and humans on two separated structures: outer and inner submucosal plexus) and the myenteric plexus located between the particular muscle layer, both the motor activity and the resorption and secretion processes occurring within the gastrointestinal tract. The aforementioned functions are carried out via the neurotransmitters synthesized and released from the neurons of the enteric nervous system [13,14,15]. The total of substances synthesized by a particular neuron is its chemical phenotype. A characteristic feature of both the central and peripheral neurons is their ability to adapt and change during different pathological conditions. This property, already observed in phylogenetically primitive species, is referred to as neural plasticity [16]. The essence of this process is to rearrange the nerve cell genome so that by changing the synthesizing neurotransmitters, it will be able to deal with the changes occurring in the surrounding environment and continue to serve its function.

The jejunum is the longest section of the digestive tract in mammals and the section where intensive nutrient absorption processes take place [17]. Therefore, disturbances in the course of this process may result in adverse systemic changes. The current study focused on studying the quantitative changes in the neurons of the enteric nervous system of the porcine jejunum. Of the numerous biologically active substances synthesized in the neuron, the current study covered substance P (SP), galanin (GAL), a neuronal form of nitric oxide synthase (nNOS), the vesicular acetylcholine transporter (VAChT) and cocaine- and amphetamine-regulated transcript (CART). SP is considered to be the major sensory neurotransmitter and/or neuromodulator involved in the emergence of pain stimuli in external and internal primary afferent neurons of the gut. Moreover, it is also responsible for blood flow regulation as well as control of gastric glands functions [18]. In turn, GAL in the gastrointestinal tract plays an important role in regulation of inflammatory processes and exhibits neuroprotective properties [19]. Nitric oxide is an unstable gaseous neurotransmitter. In the alimentary tract, nitric oxide is considered as an inhibitory factor. The inhibitory function of NO is especially prominent in reference to the smooth muscle of the jejunum, where it causes relaxation of smooth muscle and a drop in the motor functions [15,20]. VAChT is a marker of cholinergic neurons. In the gut, these neurons regulate motor activity on particular parts of the intestine [21]. CART-positive neurons are widely expressed in the gastrointestinal tract. This peptide regulates food intake, gastric blood flow and motor activity of all segments of the alimentary tract [22]. All the above mentioned substances play an important role in both the physiology and pathological conditions of the gastrointestinal tract. In contrast, little is known about the changes in the synthesis of the substances as a result of chronic exposure to acrylamide. The aim of the current study is to determine the response of neurons of the enteric nervous system of the porcine jejunum to intoxication with acrylamide administered in various doses.

In recent years, the pig has gained status as a better animal model for biomedical study rather than rats or mice, with results that can be applied to humans. The main reason is that the physiological function of specific internal organs is very similar. Particularly, the gastrointestinal tract of the pig, as an omnivorous animal, physiologically exhibits many features common to those of the human gastrointestinal tract. A similar rate of blood flow, especially in the intestine area and motor activity of particular parts of the gastrointestinal tract, including jejunum should also be mentioned [23,24]. Therefore, our results are important for human safety. Infants and toddlers are the most exposed group to acrylamide exposure. Lower body weight compared with adults, high consumption of some kinds of baby foods, and their metabolism make them more vulnerable to the effects of the contaminants. Mean acrylamide exposure in children range between 0.06 and 4.32 µg/kg bw/day depending on country of residence and eating habits [25]. Additionally, the estimated dietary intake (EDI = 2 µg/kg bw/day) was exceeded among 7% of children aged 12–36 months [26]. In the present study, we used 2 doses of acrylamide (0.5 µg/kg bw/day and 10 times higher—5 µg/kg bw/day) in young animals which are the most exposed to the negative effects of acrylamide. Doses of acrylamide used in the present study are related to human exposure to this substance and can be extrapolated to human toxicology.

## 2. Materials and Methods

### 2.1. Animals

Fifteen immature gilts at the age of 8 weeks Danish landrace were used. Animals were purchase from a local farm. At the beginning of the experiment, the body weight of pigs was 15 kg. Immediately after onset of acrylamide supplementation, pigs were divided into three groups: the control group and two experimental groups, each of them contained 5 animals (*n* = 5). Animals were kept in cages compliant with the requirements for this species with natural lighting conditions. Due to the fact that daily exposure to acrylamide contained in food products in humans ranges between 0.3 to 0.6 µg/kg of body weight [23] animals from the first experimental group receiving tolerable daily intake (TDI) dose (0.5 µg/kg bw/day); (>99%; Sigma-Aldrich, Poznań, Poland) in gelatin capsules, while gilts from second experimental group receiving a high dose of acrylamide (ten times higher than TDI, i.e., 5 µg/kg bw/day) in gelatin capsules. Pigs from the control group received empty gelatin capsules. The capsules were administered orally for a period of 28 days after the morning feeding. One time per week animals were weighed and the dose was adjusted to the actual weight of the animal. During the experiment period, all pigs received the same standardized diet (rapeseed meal 6.0%, soybean meal 9.0%, wheat 54.0%, barley 28.5%, and others 2.5%) and tap water at libitum. After the end of the experiment, all animals were euthanized with lethal doses of sodium pentobarbital (Morbital, Biowet Puławy, Puławy, Poland).

### 2.2. Tissue Collection

Then, a midline laparotomy was performed and all sections of the gastrointestinal tract were removed. Approximately 1 cm fragments of the jejunum (approximately 20 cm from the duodenum) from each animal was collected for further research. Tissue were subjected to the standard immunofluorescence frozen procedure by Makowska et al. as previously described [25]. Collected jejunum fragments were placed in a 4% buffered solution of paraformaldehyde (pH 7.4). Immersion fixation time was 1 h. Subsequently, samples were transferred to a phosphate buffer solution (PBS, pH 7.4) for 72 h (a buffer was exchanged 3 times, every 24 h). After this time, the collected jejunum were dehydrated into an 18% buffered sucrose solution for two weeks. After this time, frozen blokes were performed. The tissue blocks were cut in frontal or sagittal planes by means of a Microm HM 560 cryostat (Carl Zeiss, Berlin, Germany) at a thickness of 14 μm and attached on gelatinized glass slides suitable for immunohistochemistry.

### 2.3. Immunofluorescence Procedures

The following step was a double immunohistochemistry staining procedure that included washing in a buffer solution (PBS, 3 times, 10 min) and blocking in a blocking mixture (10% horse serum, 0.1% bovine serum albumin in 0.1 MPBS, 1% Triton X-100, 0.05% thimerosal, and 0.01% sodium aside) for 1 h. For detection of investigated substances, following unconjugated primary antibody were used. In order to investigate intraganglionic distribution of the perikarya, one of two antibodies used was directed against protein gene-product 9.5 (PGP 9.5, working dilution 1:1000, mouse, Bio-Rad, Hercules, CA, USA, code 7863-2004, used here as a pan-neuronal marker), substance P (SP, working dilution 1:150, rat monoclonal, AbD Serotec, Raleigh, NC, USA, code 8450-0505;), cocaine- and amphetamine-regulated transcript peptide (CART, working dilution 1:8000; rabbit, Phoenix Pharmaceuticals, Burlingame, CA, USA, code H-003-61), neuronal isoform of nitric oxide synthase (nNOS; working dilution 1:2000, rabbit, Sigma-Aldrich, Saint Louis, MO, USA, code AB5380), vesicular acetylcholine transporter (VAChT, working dilatation 1: 2000, rabbit Phoenix Pharmaceuticals, Burlingame, CA, USA, code H-V007) and galanin (GAL, working dilatation 1:1000, rabbit, Millipore, Billerica, MA, USA, code AB 2233). The slides were incubated overnight in a humid chamber. To visualize investigated antibody on the next day secondary antisera were used: Alexa Fluor 488 (working dilution 1:1000, donkey anti-mouse IgG, Invitrogen, Carlsbad, CA, USA, code A21202), Alexa Fluor 546 (working dilution 1:1000, goat anti-rabbit IgG, Invitrogen, Carlsbad, CA, USA, A11010), and Alexa Fluor 546 (working dilution 1:1000, donkey anti-rat, Invitrogen, Carlsbad, CA, USA, code A21208). Finally, the samples were cover slipped using a mixture of glycerol with carbonate buffer (pH = 8.4). Negative controls employed in the immunofluorescence procedure included pre-absorption test. The test was performed as follows: sections of the ganglions were incubated with “working” dilutions of primary antibodies directed toward SP, VAChT, nNOS, GAL and CART that had been preabsorbed for 18 h at 37 °C with 20 g of appropriate purified protein. Additional negative controls involved omission and replacement of all primary antisera with non-immune sera.

### 2.4. Statistical Analysis

To evaluate the percentage of exanimated neurons, at least 500 of PGP 9.5-labelled cell bodies in a definite plexus (the myenteric plexus (MP), the outer submucosal plexus (OSP) and the inner submucosal plexus (ISP)) of the studied pigs were examined. Only neurons with well-visible nucleus were counted. To prevent double counting of PGP 9.5-immunoreactive cell bodies, the sections were located at least 100 μm apart. The stained sections were analyzed under an fluorescence microscope Olympus BX 51 (Tokyo, Japan), with epi-fluorescence and appropriate filter sets, coupled with a digital camera (Olympus XM 10) connected to a PC and analyzed with Cell F software (Olympus, Tokyo, Japan).

## 3. Results

### 3.1. CART-Immunoreactive Neurons

CART-expressing neurons were most abundant in the MP: 10.90 (±0.69) %, while in the submucosal plexuses, their number was considerably smaller and amounted to 3.73 (±0.40) % in the OSP and to only 0.87 (±0.26) % in the ISP, respectively [Figure 1 and Figure 2A–I]. The rise in the number of CART-positive cell bodies in the experimental groups was determined by the acrylamide dose [Figure 1]. In the animals receiving a low dose, a statistically significant increase took place only in the neurons of the MP: 10.90 (±0.69) % [Figure 2D,G], while no changes were observed in the submucosal plexuses in the population of CART-positive neurons. In the animals administered a high dose, quantitative changes were observed in the OSP: 5.35 (±0.29) %, and a very pronounced increase was noted in the MP: 21.25 (±0.63) % [Figure 2G,H].

### 3.2. VAChT-Immunoreactive Neurons

The neurons immunoreactive for this neurotransmitter were the most abundant population, considering all the tested substances [Figure 3 and Figure 4A–I]. In the control group, a particularly large number of these neurons were observed in the submucosal plexuses: 41.24 (±1.45) % in the OSP [Figure 4B] and 39.33 (±1.43) % in the ISP [Figure 4C], respectively, while in the MP, the population of VAChT-positive neurons was 22.44 (±1.09) % [Figure 4A]. Following low-dose acrylamide administration, a statistically significant increase in the population of VAChT-positive neurons was only observed within the neurons of the MP [Figure 4A,D,G], while in the population of these neurons, no statistically significant changes were observed in the submucosal plexuses. However, the high-dose administration of acrylamide caused changes in the population of VAChT-positive neurons in all the tested plexuses of the ENS in the jejunum [Figure 4G–I]. In the MP, 32.50 (±1.37) % of VAChT-positive neurons were observed, while in the submucosal plexuses, their number was 57.33 (±1.98) % in the OSP and 48.37 (±1.38) % in the ISP, respectively.

### 3.3. nNOS-Immunoreactive Neurons

In the control group, a particularly abundant population of nNOS neurons 33.80 (±0.44) % was observed in the MP, while their number was significantly lower and amounted to 5.82 (±0.44) % in the OSP and to 7.46 (±0.69) % in the ISP [Figure 5]. In the low-dose acrylamide group, a decrease was observed in the nNOS neuron population in the ISP: 4.61 (±0.32) % [Figure 6F] and in the MP: 29.50 (±0.64) % [Figure 6D], while no changes were noted in the OSP. In contrast, in the experimental, high-dose acrylamide group, a statistically significant drop in the population of nNOS-positive neurons was observed in all the tested plexuses: in the MP to a value of 26.77 (±1.16) %, while in the submucosal plexuses to 1.22 (±0.32) % in the OSP, and 1.77 (±0.45) % in the ISP, respectively [Figure 6G–I].

### 3.4. GAL-Immunoreactive Neurons

In the control animal group, the GAL-positive neurons were a particularly abundant population within the submucosal plexus area, with their number in the OSP amounting to 37.96 (±2.63) % and in the ISP to 41.39 (±1.50) % [Figure 7 and Figure 8A–I]. However, in the MP, the number of GAL-positive neurons was only 2.75 (±0.47) %. Following the supplementation with a low dose of acrylamide, a statistically significant increase was observed in the number of GAL-positive neurons in the MP: 6.27 (±0.60) % [Figure 8D] and in the OSP: 52.20 (±1.22) % [Figure 8E], with no changes noted for the ISP. On the other hand, a high dose of acrylamide statistically significantly increased the number of GAL-positive neurons in all the tested plexuses. Within the region of the submucosal membrane plexuses, the following numbers of GAL-positive neurons were observed: 63.61 (±1.51) % in the ISP and 55.15 (±1.48) % in the OSP. However, in the MP, the number of GAL-positive neurons was 13.26 (±1.11) % [Figure 8G–I].

### 3.5. SP-Immunoreactive Neurons

SP-positive neurons in the control group were most abundant in the OSP: 25.62 (±0.95) %, while in the ISP, their number was considerably smaller and amounted to 10.14 (±0.44) %, and in the MP, this population only amounted to 0.92 (±0.22) % [Figure 9 and Figure 10A–I]. Following low-dose acrylamide administration, an increase in the population of SP-positive neurons was only observed in the MP: 1.21 (±0.14) % [Figure 10D]. However, a high dose of acrylamide increased the number of SP-expressing neurons in both the MP: 2.35 (±0.41) % [Figure 10G] and in both submucosal plexuses: in the ISP, up to 20.58 (±1.54) % [Figure 10I], and in the OSP, to 34.81 (±0.93) % [Figure 10H].

## 4. Discussion

Since confirmation of the toxic properties of acrylamide, the volume of data on the tissues that are potentially at risk of being adversely affected by this substance has been on the increase [6]. This study presents data concerning the effect of acrylamide on the expression of biologically active substances in the neurons of the enteric nervous system of the porcine jejunum. Due to the function it serves in the body, the gastrointestinal tract is particularly vulnerable to the adverse effects of substances found in the food consumed [27,28,29]. As acrylamide is formed as a by-product during the thermal processing of food, it is a compound that frequently enters the gastrointestinal tract. Of course, it is important to note that the amount of acrylamide entering the body varies and is determined by many factors. For this reason, the current study used two acrylamide doses referred to as low and high, respectively. The study focused on the effect of these doses on the quantitative changes within the population of neurons making up the enteric nervous system of the porcine jejunum. The quantitative changes in the neurons immunoreactive for CART, VAChT, nNOS, GAL, and SP were determined using the double immunofluorescence staining method. The obtained results clearly show a close correlation in the quantitative changes between the applied dose and the tested plexus. Similar relationships can be observed in other toxicological studies (supplementation of bisphenols, non-steroidal anti-inflammatory drugs) and pathological conditions that disturb the systemic metabolism (diabetes mellitus) [30,31,32,33,34,35]. The application of acrylamide at a low dose had a significantly weaker effect than the high-dose acrylamide supplementation. This is particularly evident for the external submucosal plexus, where low doses of acrylamide caused no statistically significant changes in the number of neurons expressing the tested substances. For the internal submucosal plexus, quantitative changes also only concerned the nNOS-positive and GAL-positive neurons. However, in the myenteric plexus, even low doses caused pronounced changes in the expression of the tested substances (similar to a low dose of acrylamide), with distinct changes being observed in the number of immunoreactive neuron expressions for the tested substances. It is well known that the myenteric plexus primarily serves the function of regulating the motor activity of the gastrointestinal tract [13,14]. Therefore, changes in the neurotransmitter expression in this part of the gastrointestinal tract may result in an abnormal passage of food content, which is particularly significant for the jejunum, which is primarily responsible for the intestinal absorption processes. It should be added that other toxic substances entering the gastrointestinal tract, such as bisphenol A, also cause significant changes in the expression of the same substances in the neurons of the jejunum [30]. Long-term use of aspirin also causes changes in the chemical phenotype of the neurons of the enteric nervous system of the porcine jejunum which confirms the adaptability of the enteric neurons to adverse factors and pathological conditions [36].

Moreover, the substances tested in this study are characterized by neuroprotective and anti-inflammatory properties [37,38]. Their increase may prove that, as a result of long-term exposure to acrylamide, particularly in a high dose, an inflammation of the jejunum develops. An inflammation also develops in other studies into the effects of both exogenous substances (bisphenols, non-steroidal anti-inflammatory drugs) and metabolic changes (diabetes mellitus) or damage to nerve processes (axotomy), which results in the development of inflammation [30,31,32,33,34,35,36,39]. It is expressed by, among other things, the variability in the chemical coding of enteric neurons, in particular an increase in the population of neurons immunoreactive for SP and GAL. Moreover, an increase in the expression of SP being a neurokinin of a pro-pain nature also indicates sensory changes that develop following the consumption of acrylamide, particularly in high doses [40]. The pace of changes in the expression of the studied substances also shows a decreasing trend, which was noted for the neuronal isoform of the nitric oxide synthase. Similar results were obtained for the experimentally induced hyperglycemia [40]. Nitric oxide produced within the gastrointestinal tract is largely responsible for relaxing the smooth muscle coat and regulating the lumen of blood vessels [29]. Decreasing the lumen may have an adverse effect on the motor processes and the blood flow and thus the resorption of nutrients, which is particularly important within the jejunum area. The expression of the CART peptide has been described for each section of the gastrointestinal tract in numerous animal species as well as in humans [41]. The authors’ experiments have demonstrated an increase in the number of neurons expressing it. The pace of changes in the CART expression in the course of diseases of the gastrointestinal tract (particularly the small intestine) results, as in the current study, in an increase in its expression. The function of this peptide within the gastrointestinal tract is mainly associated with the regulation of motor processes and the passage of food content. Therefore, the increase in the number of CART-positive neurons is more evidence of the toxic effect of acrylamide on the gastrointestinal tract [42,43].

## 5. Conclusions

The results obtained in this experiment complement the existing knowledge on acrylamide toxicity and confirm the neuroplasticity of the enteric nervous system. In our research, quantitative changes in neurons expressing SP, VAChT, nNOS, GAL and SP have been demonstrated. We have also shown that acrylamide, even in low doses, can have a toxic effect on enteric neurons. It should be emphasized that the tested small intestine section (jejunum) is significant from the perspective of digestion physiology, in particular, the nutrient resorption processes occurring there. All disturbances in this part of the intestine can have an adverse effect on body weight in both animals and humans. The obtained results also indicate the significant effect of neuropeptides as agents involved in the processes of the gastrointestinal tract adaptation to adverse environmental factors.

## Figures and Tables

**Figure 1 ijerph-19-14514-f001:**
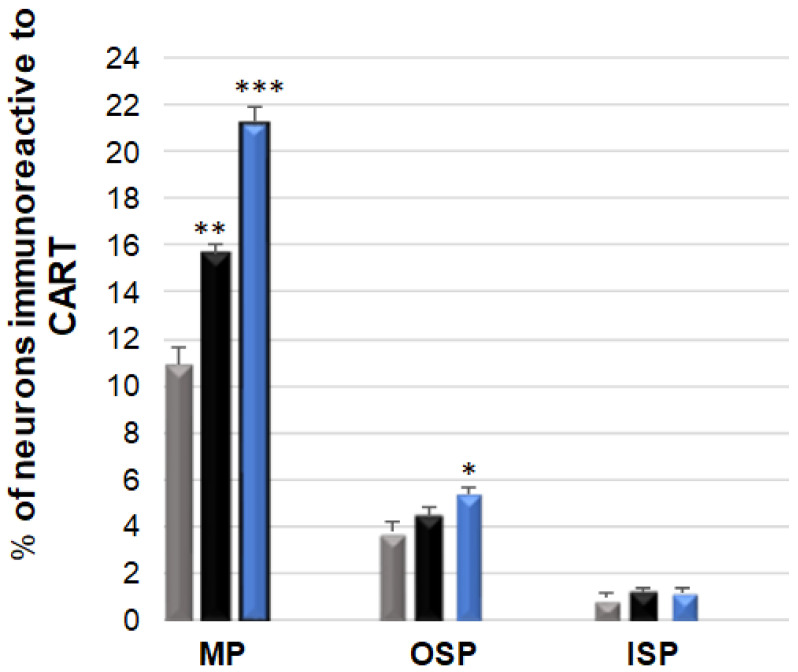
Bar graph showing the percentage variation in the number of cocaine- and amphetamine-regulated transcript (CART)—immunoreactive neurons as a result of low (black bar) and high (blue bar) dose of acrylamide supplementation in the porcine jejunum enteric plexuses. Control pigs (grey bar), MP—the myenteric plexus; OSP—the outer submucosal plexus; ISP—the inner submucosal plexus. * *p* < 0.05, ** *p* < 0.01, *** *p* < 0.001 point to differences in the expression of exact substance studied with reference to the control pigs.

**Figure 2 ijerph-19-14514-f002:**
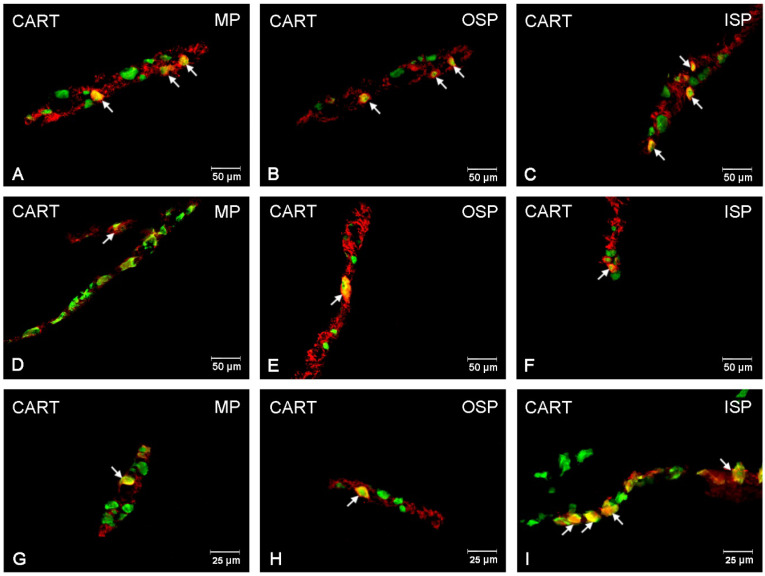
Fluorescence microscopy images showing protein gene product 9.5 (PGP 9.5) visible as green fluorescence that colocalizes with cocaine and amphetamine regulated transcript (CART) visible as red fluorescence in the enteric neurons of the jejunum in pigs. The obtained images were created by computer superimposition of two channels (green and red). The photos show the enteric nervous system neurons of the pig’s jejunum under physiological conditions (**A**,**D**,**G**) and after administration of low (**B**,**E**,**H**) and high (**C**,**F**,**I**) doses of acrylamide. Neurons immunoreactive for particular substances are advisable with arrows. MP—the myenteric plexus; OSP—the outer submucosal plexus; ISP—the inner submucosal plexus.

**Figure 3 ijerph-19-14514-f003:**
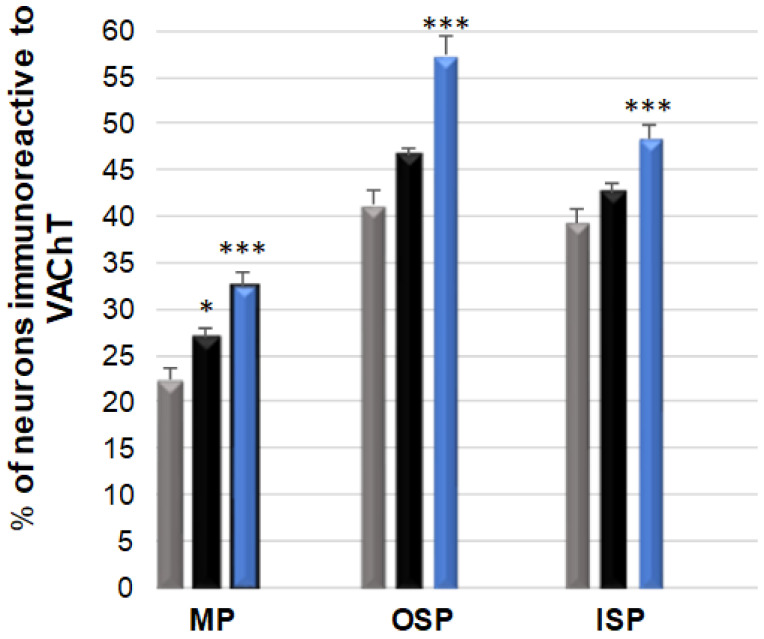
Bar graph showing the percentage variation in the number of vesicular acetylcholine transporter (VAChT) immunoreactive neurons as a result of a low (black bar) and a high (blue bar) dose of acrylamide supplementation in the porcine jejunum enteric plexuses. Control pigs (grey bar), MP—the myenteric plexus; OSP—the outer submucosal plexus; ISP—the inner submucosal plexus. * *p* < 0.05, *** *p* < 0.001 point to differences in the expression of exact substance studied with reference to the control pigs.

**Figure 4 ijerph-19-14514-f004:**
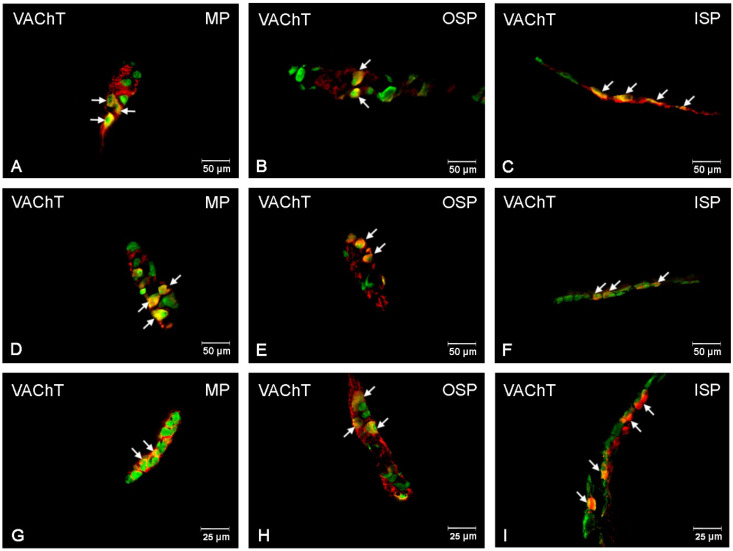
Fluorescence microscopy images showing protein gene product 9.5 (PGP 9.5) visible as green fluorescence that colocalizes with vesicular acetylcholine transporter (VAChT) visible as red fluorescence in the enteric neurons of the jejunum in pigs. The obtained images were created by computer superimposition of two channels (green and red). The photos show the enteric nervous system neurons of the pig’s jejunum under physiological conditions (**A**,**D**,**G**) and after administration of low (**B**,**E**,**H**) and high (**C**,**F**,**I**) doses of acrylamide. Neurons immunoreactive for particular substances are advisable with arrows. MP—the myenteric plexus; OSP—the outer submucosal plexus; ISP—the inner submucosal plexus.

**Figure 5 ijerph-19-14514-f005:**
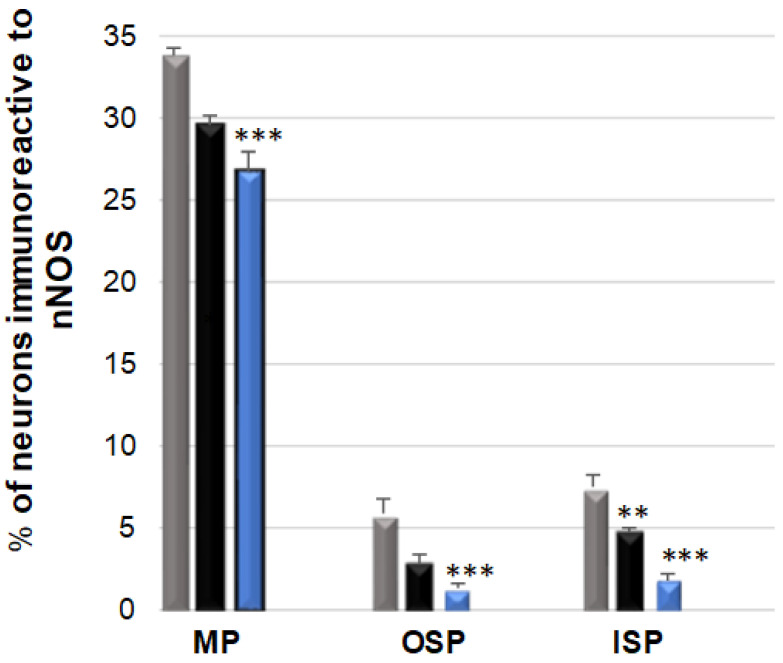
Bar graph showing the percentage variation in the number of neuronal isoform of nitric oxide synthase (nNOS)—immunoreactive neurons as a result of a low (black bar) and a high (blue bar) dose of acrylamide supplementation in the porcine jejunum enteric plexuses. Control pigs (grey bar), MP—the myenteric plexus; OSP—the outer submucosal plexus; ISP—the inner submucosal plexus. ** *p* < 0.01, *** *p* < 0.001 point to differences in the expression of exact substance studied with reference to the control pigs.

**Figure 6 ijerph-19-14514-f006:**
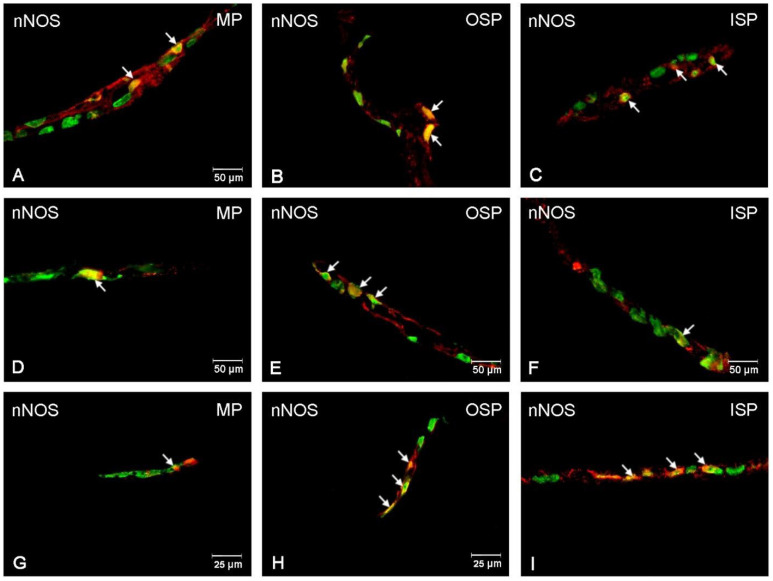
Fluorescence microscopy images showing protein gene product 9.5 (PGP 9.5) visible as green fluorescence that colocalizes with neuronal isoform of nitric oxide synthase (nNOS) visible as red fluorescence in the enteric neurons of the jejunum in pigs. The obtained images were created by computer superimposition of two channels (green and red). The photos show the enteric nervous system neurons of the pig’s jejunum under physiological conditions (**A**,**D**,**G**) and after administration of low (**B**,**E**,**H**) and high (**C**,**F**,**I**) doses of acrylamide. Neurons immunoreactive for particular substances are advisable with arrows. MP—the myenteric plexus; OSP—the outer submucosal plexus; ISP—the inner submucosal plexus.

**Figure 7 ijerph-19-14514-f007:**
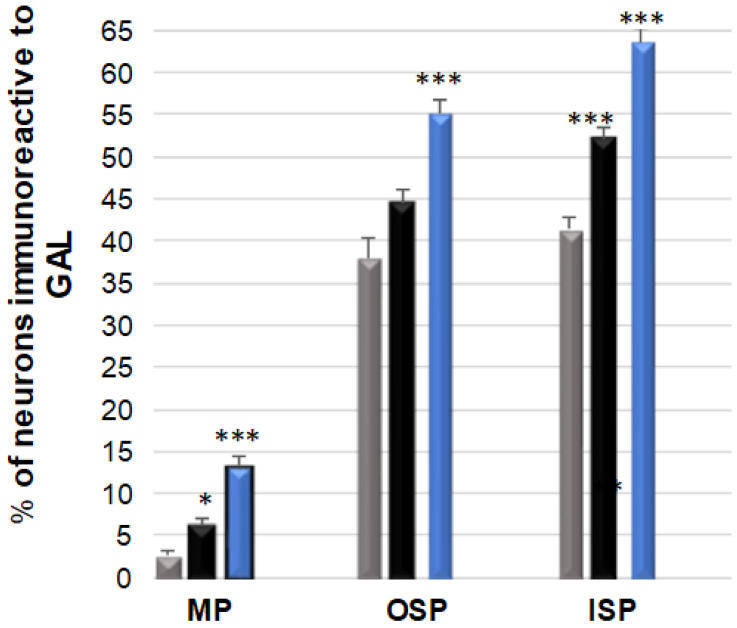
Bar graph showing the percentage variation in the number of galanin (GAL)—immunoreactive neurons as a result of a low (black bar) and a high (blue bar) dose of acrylamide supplementation in the porcine jejunum enteric plexuses. Control pigs (grey bar), MP—the myenteric plexus; OSP—the outer submucosal plexus; ISP—the inner submucosal plexus. * *p* < 0.01, *** *p* < 0.001 point to differences in the expression of exact substance studied with reference to the control pigs.

**Figure 8 ijerph-19-14514-f008:**
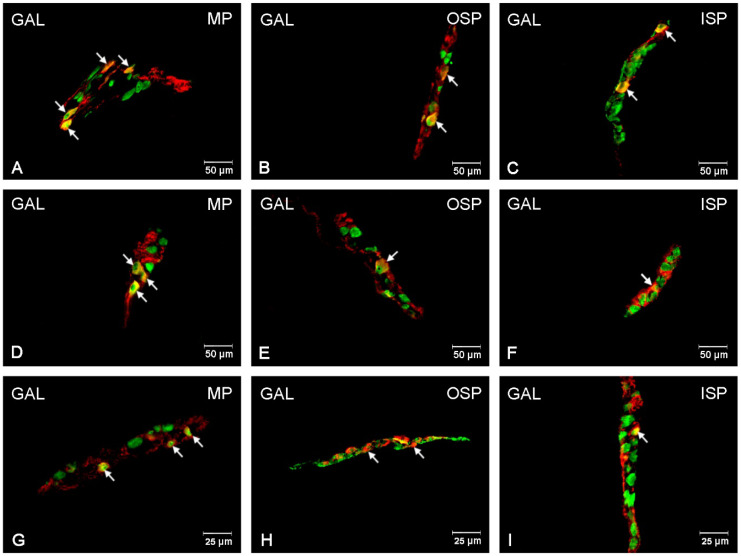
Fluorescence microscopy images showing protein gene product 9.5 (PGP 9.5) visible as green fluorescence that colocalizes with galanin (GAL) visible as red fluorescence in the enteric neurons of the jejunum in pigs. The obtained images were created by computer superimposition of two channels (green and red). The photos show the enteric nervous system neurons of the pig’s jejunum under physiological conditions (**A**,**D**,**G**) and after administration of low (**B**,**E**,**H**) and high (**C**,**F**,**I**) doses of acrylamide. Neurons immunoreactive for particular substances are advisable with arrows. MP—the myenteric plexus; OSP—the outer submucosal plexus; ISP—the inner submucosal plexus.

**Figure 9 ijerph-19-14514-f009:**
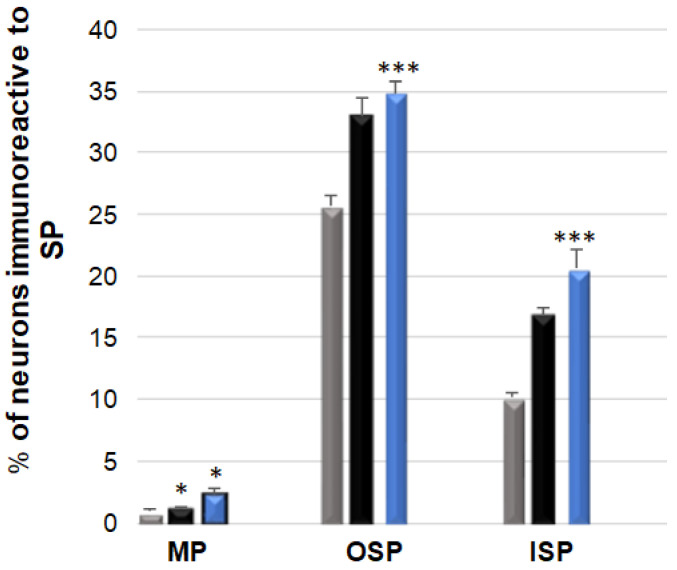
Bar graph showing the percentage variation in the number of substance P (SP)—immuno-reactive neurons as a result of low (black bar) and high (blue bar) dose of acrylamide supplementation in the porcine jejunum enteric plexuses. Control pigs (grey bar), MP—the myenteric plexus; OSP—the outer submucosal plexus; ISP—the inner submucosal plexus. * *p* < 0.05, *** *p* < 0.001 point to differences in the expression of exact substance studied with reference to the control pigs.

**Figure 10 ijerph-19-14514-f010:**
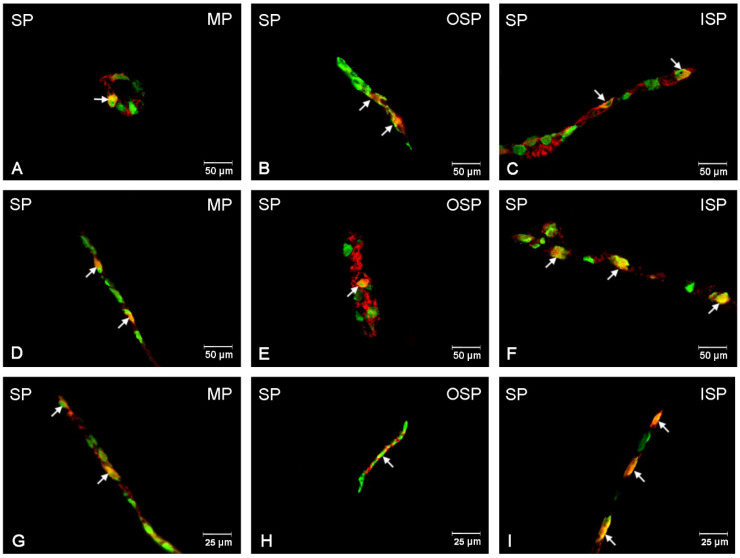
Fluorescence microscopy images showing protein gene product 9.5 (PGP 9.5) visible as green fluorescence that colocalizes with substance P (SP) visible as red fluorescence in the enteric neurons of the jejunum in pigs. The obtained images were created by computer superimposition of two channels (green and red). The photos show the enteric nervous system neurons of the pig’s jejunum under physiological conditions (**A**,**D**,**G**) and after administration of low (**B**,**E**,**H**) and high (**C**,**F**,**I**) doses of acrylamide. Neurons immunoreactive for particular substances are advisable with arrows. MP—the myenteric plexus; OSP—the outer submucosal plexus; ISP—the inner submucosal plexus.

## Data Availability

The data presented in this study are available on request from the corresponding author.

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
