# Peer review of "Administration of Different Doses of Acrylamide Changed the Chemical Coding of Enteric Neurons in the Jejunum in Gilts"

_ijerph, 2022, doi:10.3390/ijerph192114514_

Round 1
Reviewer 1 Report
The authors studied the toxicity effect of acrylamide on the enteric nervous system (ENS) of the porcine jejunum. Their results show that acrylamide causes a statistically significant increase in the number of neurons immunoreactive to SP, GAL, VAChT and CART in all types of examined enteric plexuses and a significant drop in the population of nNOS-positive enteric neurons. The results are consistent with the expected. On the other hand, the authors should pay attention that Plagiarism is about 40%. It must be decreased. Otherwise, the introduction part should be longer. The methods should be explained in more detail. And, the conclusion part is too short.
Author Response
Reviewer 1
Comments:
The authors studied the toxicity effect of acrylamide on the enteric nervous system (ENS) of the porcine jejunum. Their results show that acrylamide causes a statistically significant increase in the number of neurons immunoreactive to SP, GAL, VAChT and CART in all types of examined enteric plexuses and a significant drop in the population of nNOS-positive enteric neurons. The results are consistent with the expected. On the other hand, the authors should pay attention that Plagiarism is about 40%. It must be decreased. Otherwise, the introduction part should be longer. The methods should be explained in more detail. And, the conclusion part is too short.
*On the other hand, the authors should pay attention that Plagiarism is about 40%
Authors answer:
Thank you for your suggestion.
The authors received a plagiarism report. Most of the text that showed similarities to other articles has been removed or corrected. After improvement, only similarities concern to the names of the active substances, anatomical structures of gastrointestinal tract or the chemical reagents used in our experiment are present.
*Otherwise, the introduction part should be longer
Authors answer:
Thank you for your suggestion. The introduction section has been elongated.
*The methods should be explained in more detail
Authors answer:
Thank you for your suggestion. The methods section has been described in more detail.
* The conclusion part is too short.
Authors answer:
Thank you for your suggestion. The conclusion has been expanded.

Reviewer 2 Report
Comments to the Authors of manuscript number: ijerph-1963566 entitled “Long term administration of different doses of acrylamide changed the neurochemical phenotype of enteric neurons in the jejunum in gilts”.
The study was performed on females pigs unknown age exposed to acrylamide. Authors did not explain how dosed and time were determined. Why They used gilts. It should be explained. Moreover, the Journal relates to public health, thus it should be explained how these doses used can be extrapolated to humans.
On the other hand, the study is very important and is performed in a proper manner. It is worth to be published after proper correction.
Is long term administration performed for 28 days? Really?
1. L 10- impropriate phrase “human body”
2. L 10- exposes …to exposure – it should be rephrased
3. L 11 – please use health instead of body
4. L 35 – it is well understood
5. L 42 – do not use body, rather organism
6. L 62- 77 this part should be rephrased. The first have to presented information about the jejunum and the neurotransmitter, and then the hypothesis should be presented as well the goal of the study
7. L 79 – what does it mean :immature”? the age should be given.
8. L 84 – who did determine it?
9. L 86 – the dose given should be presented precisely.
10. all figures have to be corrected by the avoid of the description “jejunum”. This paper relates to the jejunum only.
11. too many authors' own citations
12. how the time of experiment was determined.
13. the daily consumption of acrylamide by people should be presented. This data are available.
Author Response
Reviewer 2
Comments:
Comments to the Authors of manuscript number: ijerph-1963566 entitled “Long term administration of different doses of acrylamide changed the neurochemical phenotype of enteric neurons in the jejunum in gilts”.
*The study was performed on females pigs unknown age exposed to acrylamide.
Authors answer:
Thank you for your comments. This information has been added in the material and method section.
*Authors did not explain how dosed and time were determined. Why They used gilts. It should be explained.
Authors answer:
Thank you for your comments.
Daily exposure to acrylamide contained in food products in humans ranges between 0.3 to 0.6 µg/kg of body weight [23].
In the current study, for the first time, the impact of acrylamide in tolerable daily intake (TDI) (0.5 µg/ kg of body weight (b.w.)/day) and ten times higher (5 µg/ kg of b.w./day) doses was examined, which reflects the actual intake of acrylamide in the human population.
This study was conducted on the domestic pig model, whose gastrointestinal tract is similar to humans due to its anatomy, histology and the physiological processes in it, which makes it the most suitable animal to be used in a study of the effect of food toxins on the gastrointestinal tract.
*Moreover, the Journal relates to public health, thus it should be explained how these doses used can be extrapolated to humans.
Authors answer:
Thank you for your suggestion. This information has been added to the introduction (lines 91-100).
*Is long term administration performed for 28 days? Really?
Authors answer:
The 4-week dosing period corresponds to the chronic exposure of pigs to this compound. Many previous publications on the effects of toxic substances on the gastrointestinal tract of pigs use this supplementation period (listed below) . Due to the rapid increase in weight and size of pigs during this period, it is considered optimal for the study of chronic toxicity in this species.
- The Endocrine Disruptor Bisphenol A (BPA) Affects the Enteric Neurons Immunoreactive to Neuregulin 1 (NRG1) in the Enteric Nervous System of the Porcine Large Intestine. Szymańska K, Makowska K, Całka J, Gonkowski S. Int J Mol Sci. 2020 Nov 19;21(22):8743. doi: 10.3390/ijms21228743.
- Neurochemistry of Enteric Neurons Following Prolonged Indomethacin Administration in the Porcine Duodenum. Czajkowska M, Całka J. Front Pharmacol. 2020 Sep 8;11:564457. doi: 10.3389/fphar.2020.564457.
- Prolonged acetylsalicylic-acid-supplementation-induced gastritis affects the chemical coding of the stomach innervating vagal efferent neurons in the porcine dorsal motor vagal nucleus (DMX). Gańko M, Całka J. J Mol Neurosci. 2014;54(2):188-98. doi: 10.1007/s12031-014-0274-y. Epub 2014 Mar 19.
- The Impact of T-2 Toxin on Vasoactive Intestinal Polypeptide-Like Immunoreactive (VIP-LI) Nerve Structures in the Wall of the Porcine Stomach and Duodenum. Makowska K, Obremski K, Gonkowski S. Toxins (Basel). 2018 Mar 26;10(4):138. doi: 10.3390/toxins10040138.
- The Influence of Low Doses of Zearalenone and T-2 Toxin on Calcitonin Gene Related Peptide-Like Immunoreactive (CGRP-LI) Neurons in the ENS of the Porcine Descending Colon. Makowska K, Obremski K, Zielonka L, Gonkowski S. Toxins (Basel). 2017 Mar 10;9(3):98. doi: 10.3390/toxins9030098
But for better understanding we decided to change title of article on: “Prolonged administration of different doses of acrylamide changed the chemical coding of enteric neurons in the jejunum in gilts”.
*L 10- impropriate phrase “human body”
Authors answer:
Thank you for your suggestion. It has been corrected.
*L 10- exposes …to exposure – it should be rephrased
Authors answer:
Thank you for your suggestion. It has been corrected.
* L 11 – please use health instead of body
Authors answer:
Thank you for your suggestion. It has been corrected.
*L 35 – it is well understood
Authors answer:
Thank you for your suggestion. It has been corrected.
*L 42 – do not use body, rather organism
Authors answer:
Thank you for your suggestion. It has been corrected.
*L 62- 77 this part should be rephrased. The first have to presented information about the jejunum and the neurotransmitter, and then the hypothesis should be presented as well the goal of the study
Authors answer:
Thank you for your suggestion. It has been corrected.
*L 79 – what does it mean :immature”? the age should be given.
Authors answer:
Thank you for your suggestion. It has been corrected.
The age has been added in the material and methods section (line 103).
*L 84 – who did determine it?
Authors answer:
Thank you for your comments.
Daily exposure to acrylamide contained in food products in humans ranges between 0.3 to 0.6 µg/kg of body weight [23].
In the current study, for the first time, the impact of acrylamide in tolerable daily intake (TDI) (0.5 µg/ kg of body weight (b.w.)/day) and ten times higher (5 µg/ kg of b.w./day) doses was examined, which reflects the actual intake of acrylamide in the human population.
*L 86 – the dose given should be presented precisely.
Authors answer:
Thank you for your comments. The information has been added.
* all figures have to be corrected by the avoid of the description “jejunum”. This paper relates to the jejunum only.
Authors answer:
Thank you for your suggestion. It has been corrected.
*too many authors' own citations
Authors answer:
Thank you for your suggestion. We replaced own article by other literature on this topic.
*how the time of experiment was determined.
Authors answer:
We based on previous publications describing the chronic exposure of pigs to different toxins. The 4-week dosing period corresponds to the chronic exposure of pigs to this compound. Many previous publications on the effects of toxic substances on the gastrointestinal tract of pigs use this supplementation period (listed below) . Due to the rapid increase in weight and size of pigs during this period, it is considered optimal for the study of chronic toxicity in this species.
- The Endocrine Disruptor Bisphenol A (BPA) Affects the Enteric Neurons Immunoreactive to Neuregulin 1 (NRG1) in the Enteric Nervous System of the Porcine Large Intestine. Szymańska K, Makowska K, Całka J, Gonkowski S. Int J Mol Sci. 2020 Nov 19;21(22):8743. doi: 10.3390/ijms21228743.
- Neurochemistry of Enteric Neurons Following Prolonged Indomethacin Administration in the Porcine Duodenum. Czajkowska M, Całka J. Front Pharmacol. 2020 Sep 8;11:564457. doi: 10.3389/fphar.2020.564457.
- Prolonged acetylsalicylic-acid-supplementation-induced gastritis affects the chemical coding of the stomach innervating vagal efferent neurons in the porcine dorsal motor vagal nucleus (DMX). Gańko M, Całka J. J Mol Neurosci. 2014;54(2):188-98. doi: 10.1007/s12031-014-0274-y. Epub 2014 Mar 19.
- The Impact of T-2 Toxin on Vasoactive Intestinal Polypeptide-Like Immunoreactive (VIP-LI) Nerve Structures in the Wall of the Porcine Stomach and Duodenum. Makowska K, Obremski K, Gonkowski S. Toxins (Basel). 2018 Mar 26;10(4):138. doi: 10.3390/toxins10040138.
- 4. The Influence of Low Doses of Zearalenone and T-2 Toxin on Calcitonin Gene Related Peptide-Like Immunoreactive (CGRP-LI) Neurons in the ENS of the Porcine Descending Colon. Makowska K, Obremski K, Zielonka L, Gonkowski S. Toxins (Basel). 2017 Mar 10;9(3):98. doi: 10.3390/toxins9030098
*the daily consumption of acrylamide by people should be presented. This data are available.
Authors answer:
Thank you for your comments. This information was added to material and method section (lines 108-109).

Round 2
Reviewer 2 Report
Thank Authors for the explanation point by point, but Authors did not address all comments.
The paper need further correction. Authors have to explain how the dose and age animals used correspond to children diet. Further, They have to add the acrylamide content in the diet, and explain how the period of exposure was determined. The use of the same condition from the previous studies is unreasonable because acrylamide is a quite different chemical. And finally, the exposure to acrylamide for 28 days is neither long nor prolonged. A long exposure in the case of acrylamide is approximately 1-2 years. It is a common knowledge.
On the other hand the study is very interesting.
1. Authors` answer: “Daily exposure to acrylamide contained in food products in humans ranges between 0.3 to 0.6 µg/kg of body weight [23]. “
R: It is truth, but taking into account the age of animals used, what is real exposure of human infants?
What is the relationship between these doses used in the study and acrylamide exposure in children?
Authors did not explain it.
2. Authors` answer: “The 4-week dosing period corresponds to the chronic exposure of pigs to this compound. Many previous publications on the effects of toxic substances on the gastrointestinal tract of pigs use this supplementation period (listed below) . Due to the rapid increase in weight and size of pigs during this period, it is considered optimal for the study of chronic toxicity in this species.
1. The Endocrine Disruptor Bisphenol A (BPA) Affects the Enteric Neurons Immunoreactive to Neuregulin 1 (NRG1) in the Enteric Nervous System of the Porcine Large Intestine. Szymańska K, Makowska K, Całka J, Gonkowski S. Int J Mol Sci. 2020 Nov 19;21(22):8743. doi: 10.3390/ijms21228743.
2. Neurochemistry of Enteric Neurons Following Prolonged Indomethacin Administration in the Porcine Duodenum. Czajkowska M, Całka J. Front Pharmacol. 2020 Sep 8;11:564457. doi: 10.3389/fphar.2020.564457.
3. Prolonged acetylsalicylic-acid-supplementation-induced gastritis affects the chemical coding of the stomach innervating vagal efferent neurons in the porcine dorsal motor vagal nucleus (DMX). Gańko M, Całka J. J Mol Neurosci. 2014;54(2):188-98. doi: 10.1007/s12031-014-0274-y. Epub 2014 Mar 19.
4. The Impact of T-2 Toxin on Vasoactive Intestinal Polypeptide-Like Immunoreactive (VIP-LI) Nerve Structures in the Wall of the Porcine Stomach and Duodenum. Makowska K, Obremski K, Gonkowski S. Toxins (Basel). 2018 Mar 26;10(4):138. doi: 10.3390/toxins10040138.
5. 4. The Influence of Low Doses of Zearalenone and T-2 Toxin on Calcitonin Gene Related Peptide-Like Immunoreactive (CGRP-LI) Neurons in the ENS of the Porcine Descending Colon. Makowska K, Obremski K, Zielonka L, Gonkowski S. Toxins (Basel). 2017 Mar 10;9(3):98. doi: 10.3390/toxins9030098
But for better understanding we decided to change title of article on: “Prolonged administration of different doses of acrylamide changed the chemical coding of enteric neurons in the jejunum in gilts”.”
R: As Authors wrote that They perform studies on “the effect of food toxins on the gastrointestinal tract”, and listed a few papers (above mentioned), but any one does not relate to acrylamide, which is studied very intensive on different animals models. And, in the case of acrylamide the treatment approximately one or two years is considered as a long exposure. Never 28 days. It is common knowledge. It is not also prolonged.
3. Authors also did not explain how the time was determined, because as earlier these mentioned papers did not involve acrylamide, and each toxic substance is characterized by its own specific properties, and the time exposure for mycotoxin differs from that used in the case acrylamide or bisphenol. For this reason the chose the period of 28 days has to be explained reasonably.
4. what was the content of acrylamide in the basal diet. Taking into the account the process of the food preparing for animals and the components, there in the diet is acrylamide. What was its concentration?
Author Response
Thank you for your comments and suggestions. Here are ours answers
The paper need further correction. Authors have to explain how the dose and age animals used correspond to children diet. Further, They have to add the acrylamide content in the diet, and explain how the period of exposure was determined. The use of the same condition from the previous studies is unreasonable because acrylamide is a quite different chemical. And finally, the exposure to acrylamide for 28 days is neither long nor prolonged. A long exposure in the case of acrylamide is approximately 1-2 years. It is a common knowledge.
On the other hand the study is very interesting.
- Authors` answer: “Daily exposure to acrylamide contained in food products in humans ranges between 0.3 to 0.6 µg/kg of body weight [23]. “
R: It is truth, but taking into account the age of animals used, what is real exposure of human infants? What is the relationship between these doses used in the study and acrylamide exposure in children? Authors did not explain it.
AUTHORS ANSWER
Infants and toddlers are the most exposed group to acrylamide exposure. Lower body weight compared with adults, high consumption of some kinds of baby foods, and their metabolism make them more vulnerable to the effects of the contaminants. Mean acrylamide exposure in children range between 0.06 to 4.32 µg/kg bw/day depending on country of residence and eating habits [25]. Additionally, the estimated dietary intake (EDI = 2 µg/kg bw/day) was exceeded among 7% of children aged 12–36 months [26]. In the present study we used 2 doses of acrylamide (0.5 µg/kg bw/day and 10 times higher – 5 µg/kg bw/day) in young animals which are the most exposed to the negative effects of acrylamide. Both doses can be extrapolated to the real exposure of children on this compound. We added this information to the manuscript.
- Authors` answer: “The 4-week dosing period corresponds to the chronic exposure of pigs to this compound. Many previous publications on the effects of toxic substances on the gastrointestinal tract of pigs use this supplementation period (listed below) . Due to the rapid increase in weight and size of pigs during this period, it is considered optimal for the study of chronic toxicity in this species. 1. The Endocrine Disruptor Bisphenol A (BPA) Affects the Enteric Neurons Immunoreactive to Neuregulin 1 (NRG1) in the Enteric Nervous System of the Porcine Large Intestine. SzymaÅ„ska K, Makowska K, CaÅ‚ka J, Gonkowski S. Int J Mol Sci. 2020 Nov 19;21(22):8743. doi: 10.3390/ijms21228743. 2. Neurochemistry of Enteric Neurons Following Prolonged Indomethacin Administration in the Porcine Duodenum. Czajkowska M, CaÅ‚ka J. Front Pharmacol. 2020 Sep 8;11:564457. doi: 10.3389/fphar.2020.564457. 3. Prolonged acetylsalicylic-acid-supplementation-induced gastritis affects the chemical coding of the stomach innervating vagal efferent neurons in the porcine dorsal motor vagal nucleus (DMX). GaÅ„ko M, CaÅ‚ka J. J Mol Neurosci. 2014;54(2):188-98. doi: 10.1007/s12031-014-0274-y. Epub 2014 Mar 19. 4. The Impact of T-2 Toxin on Vasoactive Intestinal Polypeptide-Like Immunoreactive (VIP-LI) Nerve Structures in the Wall of the Porcine Stomach and Duodenum. Makowska K, Obremski K, Gonkowski S. Toxins (Basel). 2018 Mar 26;10(4):138. doi: 10.3390/toxins10040138. 5. 4. The Influence of Low Doses of Zearalenone and T-2 Toxin on Calcitonin Gene Related Peptide-Like Immunoreactive (CGRP-LI) Neurons in the ENS of the Porcine Descending Colon. Makowska K, Obremski K, Zielonka L, Gonkowski S. Toxins (Basel). 2017 Mar 10;9(3):98. doi: 10.3390/toxins9030098. But for better understanding we decided to change title of article on: “Prolonged administration of different doses of acrylamide changed the chemical coding of enteric neurons in the jejunum in gilts”.”
R: As Authors wrote that They perform studies on “the effect of food toxins on the gastrointestinal tract”, and listed a few papers (above mentioned), but any one does not relate to acrylamide, which is studied very intensive on different animals models. And, in the case of acrylamide the treatment approximately one or two years is considered as a long exposure. Never 28 days. It is common knowledge. It is not also prolonged.
AUTHORS ANSWER
The main goal of our study was to established the influence of acrylamide supplementation on the enteric nervous system in the porcine jejunum. There are many publications describing the effect of acrylamide on animals organism (mainly concerning rodents). The authors used different periods of acrylamide supplementation from 5 days to several years. Comparing this to our supplementation period (28 days), the use of the term prolonged may not be appropriate and may mislead the reader, so as suggested by the reviewer, we changed the title of the article to: "Administration of different doses of acrylamide changed the chemical coding of enteric neurons in the jejunum in gilts".
- Authors also did not explain how the time was determined, because as earlier these mentioned papers did not involve acrylamide, and each toxic substance is characterized by its own specific properties, and the time exposure for mycotoxin differs from that used in the case acrylamide or bisphenol. For this reason the chose the period of 28 days has to be explained reasonably.
AUTHORS ANSWER
As we mentioned above the goal of our study was to established the influence of acrylamide supplementation on the enteric nervous system in the porcine jejunum. The study was a part of the project entitled: “Effect of different doses of acrylamide administration on the immune and nervous system of the porcine gastrointestinal tract” (Funded by KNOW (Leading National Research Centre) Scientific Consortium “Healthy Animal—Safe Food”, decision of Ministry of Science and Higher Education No. 05-1/KNOW2/2015). We conducted the research on young pigs (piglets) aged 8 weeks. These animals are treated as piglets for the entire period of supplementation (4 weeks). These animals grow rapidly and gain weight. Furthermore, we conducted a preliminary study before the experiment and established that 4 weeks period of supplementation is sufficient to induce a response of the ENS (which is also supported by earlier publications on the effects of toxins on ENS).
- what was the content of acrylamide in the basal diet. Taking into the account the process of the food preparing for animals and the components, there in the diet is acrylamide. What was its concentration?
AUTHORS ANSWER
Pigs used in the experiment received a feed that was not heat treated above 100 ° C, which is necessary for the formation of acrylamide. A very high level of acrylamide is found in potato chips, French fries, corn flakes, crackers, and coffee. Some authors argue that animals may also be exposed to acrylamide. Partly, food by-products and faulty bakery products are used for animal feeding, too. Possibly, pelletizing during the processing of mixed concentrates may also be involved in acrylamide formation. However, data on their levels in animal feed are not available.